# Integrating [^18^F]-Fluorodeoxyglucose Positron Emission Tomography with Computed Tomography with Radiation Therapy and Immunomodulation in Precision Therapy for Solid Tumors

**DOI:** 10.3390/cancers15215179

**Published:** 2023-10-27

**Authors:** Conor M. Prendergast, Egesta Lopci, Romain-David Seban, Dorine De Jong, Samy Ammari, Sanjay Aneja, Antonin Lévy, Abin Sajan, Mary M. Salvatore, Kathleen M. Cappacione, Lawrence H. Schwartz, Eric Deutsch, Laurent Dercle

**Affiliations:** 1Department of Radiology, NewYork-Presbyterian, Columbia University Irving Medical Center, New York, NY 10032, USAms5680@cumc.columbia.edu (M.M.S.); kmc2113@cumc.columbia.edu (K.M.C.);; 2Nuclear Medicine Unit, IRCCS—Humanitas Research Hospital, 20089 Rozzano, Italy; 3Department of Nuclear Medicine, Institut Curie, 92210 Saint-Cloud, France; 4Laboratory of Translational Imaging in Oncology, Inserm, Institut Curie, 91401 Orsay, France; 5RefleXion Medical, Inc., Hayward, CA 94545, USA; 6Center for Cell Engineering, Memorial Sloan Kettering Cancer Center, New York, NY 10065, USA; 7Department of Medical Imaging, Institut Gustave Roussy, 94805 Villejuif, France; 8Department of Radiation Oncology, Smilow Cancer Hospital, Yale School of Medicine, New Haven, CT 06519, USA; 9Department of Radiation Oncology, Gustave Roussy, 94805 Villejuif, France

**Keywords:** PET imaging, abscopal effect, PET/CT, immunotherapy, radiation therapy, tumor metastasis, tumor hyperprogression, tumor biomarkers

## Abstract

**Simple Summary:**

Radiation therapy has long been reported to affect tumors distal to the site of irradiation, in what is known as the abscopal effect; the synergy of radiation with immune-oncological agents has also been studied and exploited for greater anti-cancer effect. We believe that PET/CT imaging can offer insight into the mechanism of this synergy and thereby optimize the dosing and timing as well as monitoring the response to and adverse effects of radiation therapy in tandem with immunotherapy. Herein, we offer a commentary to better integrate PET/CT into recently released joint guidelines, to exploit radiation and immunotherapy synergy.

**Abstract:**

[^18^F]-FDG positron emission tomography with computed tomography (PET/CT) imaging is widely used to enhance the quality of care in patients diagnosed with cancer. Furthermore, it holds the potential to offer insight into the synergic effect of combining radiation therapy (RT) with immuno-oncological (IO) agents. This is achieved by evaluating treatment responses both at the RT and distant tumor sites, thereby encompassing the phenomenon known as the abscopal effect. In this context, PET/CT can play an important role in establishing timelines for RT/IO administration and monitoring responses, including novel patterns such as hyperprogression, oligoprogression, and pseudoprogression, as well as immune-related adverse events. In this commentary, we explore the incremental value of PET/CT to enhance the combination of RT with IO in precision therapy for solid tumors, by offering supplementary insights to recently released joint guidelines.

## 1. Introduction

[^18^F]-fluorodeoxyglucose positron emission tomography combined with computed tomography (CT) imaging ([^18^F]-FDG PET/CT) is a pivotal tool for a wide range of indications in cancer imaging, including diagnosis, staging, prognostication, response assessment, and detection of recurrence/relapse. In the field of radiation therapy (RT), PET/CT plays a critical role in the clinical workflow, both for treatment planning and delivery.

The widespread adoption of immuno-oncological (IO) agents for the treatment of various cancers has created a paradigm shift by redefining the role of RT combined with IO, aiming to produce a synergic action against tumors. Thanks to the combination of functional and metabolic data it provides, the PET/CT modality can be a valuable support to clinicians in guiding, monitoring, and adjusting the synergic activity of RT/IO treatment plans.

In this commentary, we discuss the application of RT in the context of immunomodulatory treatments, by adapting [^18^F]-FDG-PET/CT imaging from the recently published joint guidelines on its recommended use in patients with solid tumors [1]. Our objective is to advocate for the utilization of PET/CT to monitor tumor responses to therapy, identify immune-related adverse events, and enhance existing guidelines that define PET/CT’s role in optimizing the timing of RT and IO agent administration. This aims to enhance patient outcomes and capitalize on the potential benefits of the abscopal effect.

## 2. Imaging Biomarkers Derived from Baseline PET

### 2.1. [^18^F]-FDG PET

Glucose metabolism is most often used as a surrogate of tumor metabolism for staging and prognosis. However, high baseline glucose consumption in non-tumoral hematopoietic tissue could be envisioned as a biomarker associated with systemic immunosuppression, providing clinical guidance on additional therapeutic regimens [2]. On a molecular level, such increased glucose metabolism—known as the “Warburg effect” [3] or aerobic glycolysis—is well detected by [^18^F]-FDG PET/CT. Metabolic parameters such as maximum standardized uptake values (SUVmax), mean standardized uptake values (SUVmean), metabolic tumor volume (MTV), and total lesion glycolysis (TLG) have been demonstrated as important prognostic and predictive factors in several malignancies after surgery, radiotherapy, or chemotherapy. Imaging metrics quantifying the glucose metabolism of non-tumoral hematopoietic tissue could also predict the efficacy of immunotherapy and potentially offer a novel method for stratifying and selecting appropriate patients who would most benefit from IO regimens [2]. It seems that there is a close relationship between bone marrow glucose metabolism and myeloid infiltration in the tumor microenvironment, also known to be associated with unfavorable outcomes regarding responses to IO and radiotherapy [4]. Additionally, myeloid cells in the tumor microenvironment must compete for oxygen and nutrients with the highly demanding cancer cells and adapt their energy signatures to survive. Their presence in the tumor microenvironment seems to obstruct anti-tumor T cell immunity, making them more resistant to checkpoint inhibitors [4].

### 2.2. Immuno-PET

Current imaging techniques have focused mainly on tumor cells. However, their immune context may have interesting clinical implications. Technological improvements and new radiopharmaceuticals targeting elements of the immune system and immune checkpoints have led to the progressive implementation of immuno-PET into the clinical scenario. This technology can help determine density, composition, functional state, and leukocyte infiltrate in the tumor microenvironment. Initial research has shown that the immune context can predict the efficacy of immunotherapy and overall prognosis, although this has yet to be fully confirmed in clinical trials [5]. Radiotracers targeting small proteins of checkpoints PD-1/PD-(L)1 (lymphocytic exhaustion), tumor-infiltrating lymphocytes such as CD3 and CD8 (cytotoxic lymphocytes), enzymes (ex. Granzyme B, dCK deoxycytidine kinase, dGK deoxyguanosine kinase), [^18^F]-F-AraG, etc., have already been studied [2,6,7,8,9]. Initial evidence in humans has proved the capability of immune-PET to non-invasively identify the expression of checkpoint inhibitors (ICIs), evaluate the immune infiltrate and its activity, and predict the response to ICI as well as overall survival. The use of dedicated tracers such as [^89^Zr]-labeled PD-1 inhibitor antibodies has helped elucidate PD-1 expression, by showing biodistribution and tumor uptake, thereby improving the prediction of RT/IO response [7]. In the future, the optimal strategy may shift from non-immune-specific imaging biomarkers to immune-specific biomarkers with immuno-PET.

## 3. The New Role of PET for Defining the Effect of IO/RT

[^18^F]-FDG PET imaging has the advantage of providing metabolic data, including the quantified tumor glucose uptake, with several potential implications in the investigation of IO/RT. These metrics help quantify tumor metabolic activity, improve histologic definition, and may provide insight on IO/RT outcomes. Metabolic information can be used to evaluate and compare responses to different strategies and sites, such as irradiated versus non-irradiated tumors [10].

### Understanding the Abscopal Effect

The abscopal effect describes a systemic effect of RT defined as the regression of metastatic tumors located at significant distance from the irradiated sites. It has been observed in multiple types of cancers and is believed to derive from immune-mediated component activation following cell death by radiation [2,4]. Radiotherapy induces cytokine release by local cells, the formation of tumor-derived antigens, increased recruitment of antigen-presenting cells, and upregulation of antigen receptors in immune cells. This summates to activation of cytotoxic T cells, the circulation of which might be responsible for distal tumor regression. Several studies have attempted to combine IO with RT to investigate a potential advantage of the radiation-induced immune response for potential anti-tumor synergy. Data derived from small trials have reported that the abscopal effect appears in up to 30% of enrolled patients [11].

To better understand the potential role of PET in defining the abscopal effect, it is important to understand the current data for the mechanisms explaining the efficacy of RT in combination with immunotherapy. Two main agents are currently used. Programmed death protein/ligand 1 (PD-L1) is expressed at the surface of several types of cancer cells; it interacts with T cells, inhibiting their proliferation and aiding in the evasion of the immune system. Blocking this pathway restores T cell efficacy and mediates anti-tumor responses. Cytotoxic T-lymphocyte antigen 4 (CTLA-4), which is expressed on T cells, binds to the B7 protein on antigen-presenting cells to stop T cell activation. Blocking this pathway with an anti-CTLA-4 antibody restores T cell activity, leading to anti-tumor responses.

In mouse models of renal, pancreatic, and melanoma cancers, daily PD-1 inhibitors and RT at 1 and 14 days have slowed tumor growth; mathematical models showed that a delay of >7 days between RT and IO was enough to significantly attenuate the radiosensitive effect [12]. However, radiotherapy with CTLA-4 and PD-1 inhibitors did not improve response in PD-1-resistant non-small-cell lung cancer [5]. In clinical trials, CTLA-4 inhibitors given before or with RT did not improve brain metastases; this may be due to the brain’s unique role as an immune-privileged site; however, in advanced-stage metastatic melanoma, non-brain RT in conjunction with CTLA-4 inhibitors significantly improved survival time [6].

The timing of RT with IO varies drastically depending on the cancer and IO agent used in mouse models. In melanoma, the dosage of CTLA-4 blockers within 4 weeks of RT diminished lesions the most. Colorectal cancer and mammary tumors had significantly higher cure rates and survival times when RT was given closer to CTLA-4 inhibitor dose dates [10]. Human data are scarcer for combinations of PD-1 inhibitors and RT, with pre-clinical studies showing optimal administration of PD-1 inhibitors within approximately 7 days of RT [13].

The delicate IO dosing window may be due to CD8+ cell levels, which peak at 5–8 days following RT in mouse models, indicating a potential window for enhanced IO effectivity [13]. In case reports published since 1969, the median time for the abscopal effect to manifest was 2 months after treatment. Shrinkage of tumors distal to the RT site also followed a similar trend, with the tumors decreasing in 1–3 months following treatment [13]. Still, the effect of clinical responses remains elusive. Metabolic changes detected by PET (Table 1), including decreased [^18^F]-FDG uptake and total lesion glycolysis, are more objective and occur earlier in time compared to morphological changes.

As for other indications, we assume that the future role of [^18^F]-FDG PET for the evaluation of the abscopal effect and RT/IO strategies will be crucial for establishing baseline metabolic information and monitoring changes (Figure 1), offering better insight into prognoses of metastases with IO/RT and helping to detect adverse effects [4].

Our proposal for using [^18^F]-FDG PET/CT to assess the abscopal effect is as follows: A bone marrow biopsy, organ evaluation, and brain MRI must be performed to provide a baseline, to be referred to in case of adverse effects of IO administration. A standard low-dose CT with or followed by PET is then performed to generate baseline data for tumor anatomical location, metabolic activity, and burden; these data will also be used to create a personalized RT, to irradiate the most metabolically active regions and spare healthy tissues. Corresponding IO agents are administered; within 7 days, RT is administered, to take advantage of the resultant window of radio sensitivity. A clinical follow-up to assess response and re-staging is performed 6–12 weeks after the RT; if the metastases distal to the primary tumor irradiation site have regressed, the abscopal effect has occurred. Based on response criteria or adverse effects, clinicians can adjust IO or RT to each patient and repeat as necessary.

## 4. The Role of PET for Refining Patterns of Response to IO/RT

### 4.1. Role of PET for Hyperprogression

The concept of hyperprogression is not well understood or quantified; it may be the result of tumor resistance to radiation or IO agents such as PD-1 inhibitors [2,14]. It poses a diagnostic dilemma to the wait-and-see strategy currently used to differentiate true progression from pseudoprogression (Figure 2). Hyperprogression has been more frequently observed in patients with a higher baseline tumor burden and multiple metastases. Further complications to the management paradigm are the wide range of hyperprogression frequencies, ranging from 8% to 30%, and the new concepts in RT, such as oligoprogression, discussed in the next section [2,14].

A potential reason for this wide frequency range is the variability in the definition criteria [1]. On one side, hyperprogression is categorized as a fast tumor progression independent of IO therapy [14]. Herein, tumor growth rate is only considered after beginning IO. This makes for a more convenient prognostic tool, as it only requires two response assessments, but this definition cannot demonstrate a causal effect: the fast progression cannot be attributed specifically to IO. Patients with a high baseline tumor burden at the beginning are more likely to fit hyperprogression criteria if only those two points in time are considered. Another definition of hyperprogression is accelerated growth as a harmful effect of IO agents. This takes into account the change between pretreatment and on-treatment tumor growth rates. This definition has a demonstrably lower rate of hyperprogression using PET. It also assumes medical imaging is available before and during IO therapy, which is often the case in patients treated with IO.

Integrating the criteria for defining hyperprogression is critical, given its importance in prognosis. To do so, it is important to take into account clinical, radiologic, and metabolic biomarkers. It is also important to differentiate fast and accelerated progression criteria and harmonize their detection methods as well as the criteria for measuring target lesions, and to consider new methods altogether.

The future role of PET in patients with solid tumors treated with a combination of IO and RT might also involve predicting patients that are more likely to become hyperprogressors based upon the extraction of biomarkers in baseline imaging [14].

### 4.2. Role of PET for Oligoprogression

Oligoprogression refers to the progression of only a few lesions in metastatic cancer; a limited number of lesions may progress even as the others regress or remain stable. This is often due to the development of resistance to targeted therapies and IO agents, and it poses a challenge for treatment [15,16]. The patient’s tumor burden is lower than that of hyperprogression, presenting fewer and smaller metastases; however, similar to hyperprogression, oligoprogression may still warrant more aggressive treatments such as local ablation, brachytherapy, or surgery where appropriate [16]. As with hyperprogression, PET/CT provides valuable insight into the detection of oligoprogression patterns. As the increase in tumor size is often smaller than with hyperprogression, more specific radiotracers with a higher sensitivity would detect these slight changes and allow for quicker interventions and better survival rates [16].

### 4.3. Role of PET for Pseudoprogression

The current response criteria for IO focus principally on the detection of the pseudoprogression phenomenon, which is defined as a brief increase in tumor burden or new tumors, followed by an abrupt decrease or stability. It is believed that IO agents may cause a brief uptick in tumor metabolism and vasculature due to inflammation, or possible resistance to the IO agent. This brief increase in size requires a follow-up scan within 4–8 weeks [1].

The incidence of pseudoprogression varies with the type of tumor and IO agent administered, reaching up to 17.9% in “progressive” patients treated with the PD-1 blocker pembrolizumab [16]. Data suggest that the outcomes of pseudoprogressive patients resemble those of treatment-sensitive patients instead of true progressive patients. This is the reason why available criteria aim to provide clinical guidance to ensure pseudoprogression is not misdiagnosed as true progression.

According to the joint guidelines [1], [^18^F]-FDG PET imaging is indicated prior to immunotherapy (due to its ability to predict and detect tumor prognosis through mean tumor volume [13]) and for response assessment [1]. This allows for the detection of different patterns of response, including pseudoprogression, which may be due to a delayed immune system activation, local inflammation, and/or immune system tumor infiltration [17]. In the context of RT/IO, [^18^F]-FDG PET can be used to dose-paint or image the heterogeneity of a tumor’s activity, though it is unclear if it benefits survival [14].

### 4.4. Role of PET for the Diagnosis of Immune-Related Adverse Events

Immunogenic or toxic effects due to RT/IO can be observed weeks or months after treatment. Radiotoxic effects may take years to develop in prolonged treatment. This makes their combined effects difficult to quantify and emphasizes the importance of investigating new biomarkers. For example, the inflammatory process induced by immune-related adverse effects (irAEs) is associated with a markedly increased [^18^F]-FDG uptake. This modality has been used to predict thyroiditis with hypothyroidism and irAE in cancer patients receiving a combination of two ICIs, even prior to the appearance of the usual clinical and biological indicators required for diagnosis of most irAEs [2]. The substrate for irAEs is the activation of the T cell inflammatory response and cytokine release [12]. Symptoms are systemic and potentially life-threatening, with the capability to occur in nearly every organ [1]. Fortunately, patients respond to steroids or discontinuation of IO therapy.

Imaging is able to detect immunogenic lesions indicative of irAE in patients 75% of the time. Thanks to its ability to depict inflammation in a whole-body modality, [^18^F]-FDG PET/CT represents the optimal tool for the early detection of irAEs [17].

## 5. Documentation and Reporting: Adaptation of the Joint Guidelines

The recently published joint EANM/SNMMI/ANZSNM practice guidelines/procedure standards on [^18^F]-FDG PET/CT use in immunomodulatory treatments [1] provide dedicated insight on how to perform, interpret, and report the scan during IO regimens [18,19].

The clinical history of the patient should be briefly summarized, including relevant diagnostic tests, prior imaging findings, and the specific type, site, timing, and number of RT/IO. Drugs potentially impacting [^18^F]-FDG uptake should be listed. Sufficient details should be recorded for follow-up imaging to be replicated or at least comparable. 

Imaging findings should be described in a logical manner; they may be grouped by significance, TNM staging, or body region. Relevant [^18^F]-FDG findings require a detailed description of the location, extent, and intensity of [^18^F]-FDG uptake with noteworthy anatomical findings in CT. Target lesions should be identified using the indications of the selected metabolic response criteria (Table 2), as well as the pattern of response used for tumor assessment [20].

There should also be a focus on significant clinical findings with respect to any additional imaging needed to clarify the impression. Any relevant abnormalities must be promptly communicated, in order to avoid treatment delays that might result in significant morbidity or mortality. It should be recognized that the use of PET may alter the rate of detecting irAEs compared to those previously identified in routine clinical care and may impact both subsequent treatments.

## 6. Conclusions

Recent advances in solid tumor treatment with RT combined with IO treatment require advanced imaging and analysis to determine real-time benefits and complications to novel interventions. The importance of [^18^F]-FDG PET/CT for the evaluation of the abscopal effect [28] and RT/IO strategies [28,29] will be crucial in establishing baseline metabolic information and providing increased insight to response, as well as for the detection of adverse treatment effects. Still, the modality has its limitations. [^18^F]-FDG uptake is dependent on various tumor factors (size, metabolic activity, and serum glucose), which can lead to false negatives in very small lesions (for example, micro-metastases), and tumors with a lower metabolic rate (prostate, hepatic, etc.). In the future, we can expect [^18^F]-FDG PET to be supplemented by novel radio tracers and molecule imaging, including immune-PET, radiolabelled antibodies, and agents with more exclusive expression in cancerous lesions, improving specificity and yielding higher tumor-to-normal ratios. Moreover, the implementation of artificial intelligence (AI) and dedicated algorithms for image analysis and the evaluation of follow-up studies will further assist therapy monitoring using PET and help determine RT/IO timing and dosage [2,30].

## Figures and Tables

**Figure 1 cancers-15-05179-f001:**
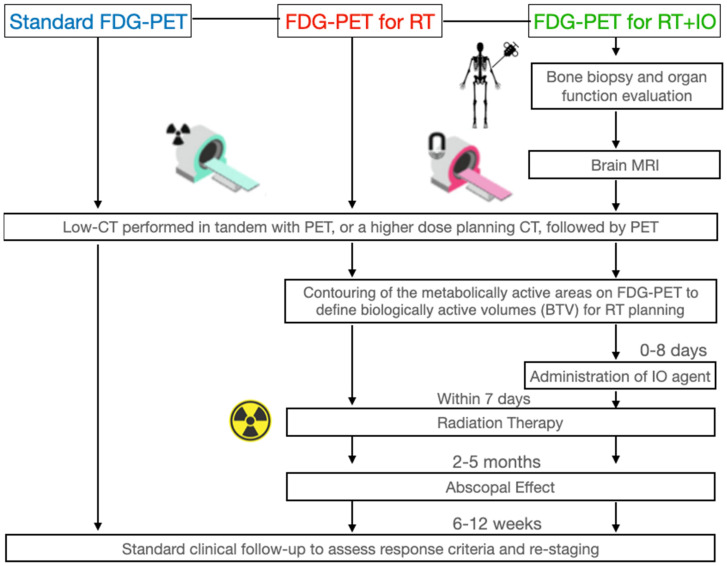
Guidelines on [^18^F]-FDG PET for radiotherapy and radiotherapy + immuno-oncology, in comparison to standard [^18^F]-FDG PET. Bone marrow biopsy, organ function assessment, and brain MRI are performed when immuno-oncological agents are added, to provide a baseline in case of immune-related adverse events during treatment. We recommend a clinical and imaging follow-up after 5 months to assess primary and metastatic response. Regression of tumors distant to the primary could indicate the abscopal effect.

**Figure 2 cancers-15-05179-f002:**
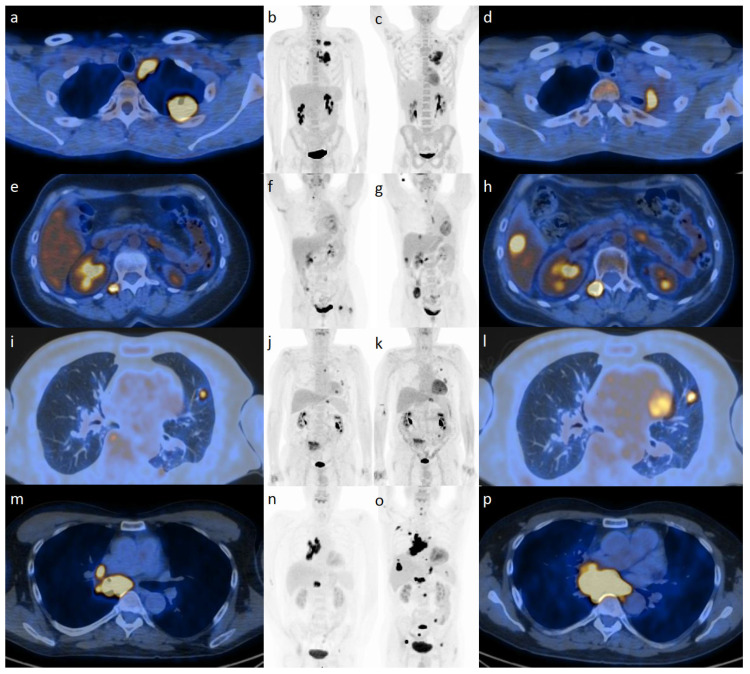
Patterns of response from [^18^F]-FDG PET during immunotherapy with checkpoint inhibitors illustrated at baseline (**a**,**b**,**e**,**f**,**i**,**j**,**m**,**n**) and during treatment (**c**,**d**,**g**,**h**,**k**,**l**,**o**,**p**): (starting from upper panels) partial response (**a**–**d**), progressive disease (**e**–**h**), pseudoprogression (**i**–**l**), and hyperprogression (**m**–**p**).

**Table 1 cancers-15-05179-t001:** Overview of typical PET parameters: significant and measured metrics.

PET Parameters	Point of Interest
Standardized uptake value (SUV)	Dimensionless value of tumor’s’ uptake of the radiotracer, quantified as SUVmax, SUVmean, SUVpeak; helps distinguish between metabolically normal and abnormal tissues
Metabolically active tumor volume (MATV)	Volume of [^18^F]-FDG-avid tumor delineated on PET images; utilized for prognosis and quantifying response to treatment
Total lesion glycolysis (TLG)	Product of the mean standardized uptake value and metabolic activity tumor volume; used to quantify metabolic tumor burden
Bone marrow avidity	Metric of [^18^F]-FDG uptake in bone marrow; often higher than other organs such as the liver and could be a biomarker for detecting immunosuppressive cancers or infiltration by cancers such as lymphomas
Bone marrow to liver ratio/spleen to liver ratio	Ratio between the [^18^F]-FDG uptake in bone marrow or in the spleen compared to the liver uptake; used to assess the immune system activation in the context of IO

**Table 2 cancers-15-05179-t002:** Immune-related response criteria. Modified from Lopci et al. [1] under a Creative Commons Attribution 4.0 International License (http://creativecommons.org/licenses/by/4.0/) URL (accessed on 31 July 2023).

	EORTC	PERCIST 1.0	LYRIC	PECRIT	PERCIMT	imPERCIST	iPERCIST
Authors	Young et al. [21]	Wahl et al. [22]	Cheson et al. [23]	Cho et al. [24]	Anwar et al. [25]	Ito et al. [26]	Goldfarb et al. [27]
Tumor type/Modality	Solid tumor/[^18^F]-FDG PET	Solid tumor/[^18^F]-FDG PET	Lymphoma/CT and [^18^F]-FDG PET	Melanoma/CT and [^18^F]-FDG PET/CT	CT and [^18^F]-FDG PET/CT	[^18^F]-FDG PET/CT	[^18^F]-FDG PET/CT
Year	1999	2009	2016	2017	2017	2019	2019
Lesion measurement	-	-	Bidimensional	Unidimensional	-	-	-
Baseline size	-	-	>15 mm	>10 mm	-	-	-
Baseline lesion number	-	5 lesions total, 2 per organ	6 lesions total (nodes and extranodal sites)	5 lesions (2 per organ)	According to RECIST 1.1 and PERCIST 1	5 lesions total, 2 per organ	5 lesions total, 2 per organ
New lesion	Results in PD	Results in PMD	Considered as IR2a	Results in PD	PD depends on number and functional size of new lesion(s)	SULpeak of new lesion(s) included in the sum of SULpeak	To be confirmed by a new imaging evaluation at least 4 weeks later
Non-index lesion	-	-	Considered as IR2b	Same as RECIST 1.1	-	-	-
Complete resolution	Complete resolution of [^18^F]-FDG PET uptake within the tumor volume so that it is indistinguishable from surrounding normal tissue	Disappearance of all metabolically active lesions	[^18^F]-FDG PET-uptake < liver (score 1, 2, 3) without a residual mass OR on CT, target nodes/nodal masses must regress to <15 mm in longest diameter	See RECIST 1.1 Results in clinical benefit	No new lesions	Disappearance of all lesions	Disappearance of all lesions
Partial reduction	A reduction of a minimum of 15–25% in tumor SUV after one cycle of chemotherapy, and >25% after more than one treatment cycle	Reduction in SULpeak in target lesions of > 30% and absolute drop in SUL > 0.8 SUL units	[^18^F]-FDG PET-uptake > liver (score 4 or 5) with reduced uptake compared with baseline and residual masse(es) of any size OR on CT > 50% decrease in SPDof up to 6 measurable nodes and extranodal sites	See RECIST 1.1 Results in clinical benefit	No new lesions	≥30% decrease in sum of SULpeak of target lesions and decrease of ≥0.8 SUL units	≥30% decrease in sum of SULpeak
Stable disease	An increase in SUV < 25% or a decrease < 15% and no visible increase in extent of [^18^F]-FDG PET tumor uptake (>20% in the longest dimension)	Neither CR/PR nor PD can be established	Neither CR/PR nor PD can be established	See RECIST 1.1 If SULpeak decreases by more than 15.5%, clinical benefit If SULpeak decreases by less than 15.5%, no clinical benefit	Neither CR/PR nor PD can be established	Neither CR/PR nor PD can be established	Neither CR/PR nor PD can be established
Progressive disease	An increase in SUV > 25% within the tumor region defined on the baseline scan, visible increase in the extent [^18^F]-FDG PET tumor uptake (>20% in the longest dimension) or the appearance of new [^18^F]-FDG PET uptake in metastatic lesions	Increase in SULpeak of > 30% or the appearance of a new lesion	First PD is IR (indeterminate response) Increase > 5 mm (if <2 cm) or 10 mm (if >2 cm) of at least one lesion Criteria for IR IR1: >50% increase in SPD in first 12 weeks IR2a: <50% increase in SPD with new lesion IR2b: <50% increase in SPD with >50% increase in product of the perpendicular diameters of a lesion or set of lesions IR3: increase in [^18^F]-FDG PET uptake without a concomitant increase in lesion size meeting criteria for IR1 or IR2	See RECIST 1.1 Results in no clinical benefit	Four or more new lesions of less than 1.0 cm in functional diameter, or three or more new lesions of more than 1.0 cm in functional diameter, or two or more new lesions of more than 1.5 cm in functional diameter Predicts clinical PD, and no clinical benefit	>30% increase in sum of SULpeak	>30% increase in sum of SULpeak or new lesions Results in UPMD Clinical stability is considered to decide if treatment should be continued after UPMD
Confirmation PD	No	No	Yes, wait up to 12 weeks	No	No	No	Yes, 4–8 weeks later for CPMD

## Data Availability

The data presented in this study are available in this article.

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
