# Peer review of "Integrating [18F]-Fluorodeoxyglucose Positron Emission Tomography with Computed Tomography with Radiation Therapy and Immunomodulation in Precision Therapy for Solid Tumors"

_cancers, 2023, doi:10.3390/cancers15215179_

Round 1

Reviewer 1 Report (New Reviewer)

The manuscript provides an excellent literature review, focusing on the crucial discussions concerning PET CT and its evaluation of response. The authors have done an admirable job in covering the essential aspects of immuno PET and confounding factors. Despite the information being available in the public domain, the authors have skillfully presented a comprehensive and coherent picture.

To further enhance the manuscript's impact, I recommend including a selection of representative images for different cases, with at least one in each category. The addition of these interesting visuals will substantially elevate the overall presentation, preventing the manuscript from becoming monotonous.

Author Response

Reviewer 1:

The manuscript provides an excellent literature review, focusing on the crucial discussions concerning PET CT and its evaluation of response. The authors have done an admirable job in covering the essential aspects of immuno PET and confounding factors. Despite the information being available in the public domain, the authors have skillfully presented a comprehensive and coherent picture.

To further enhance the manuscript's impact, I recommend including a selection of representative images for different cases, with at least one in each category. The addition of these interesting visuals will substantially elevate the overall presentation, preventing the manuscript from becoming monotonous.

Answer: We thank you for your time and comments. We have included images to represent each response and enhance the engagement of the paper.

Reviewer 2 Report (Previous Reviewer 2)

I believe that, after these revisions, the important citations have been added. Additionally, upon further review, I had not initially noticed that this was a commentary. Now, I consider this article suitable for publication in Cancers. Thank you to the authors for their diligent and responsible approach.

Author Response

Reviewer 2:

I believe that, after these revisions, the important citations have been added. Additionally, upon further review, I had not initially noticed that this was a commentary. Now, I consider this article suitable for publication in Cancers. Thank you to the authors for their diligent and responsible approach.

Answer: We thank you for your time and comments.

Reviewer 3 Report (Previous Reviewer 3)

The manuscript has shown improvements. However, several obvious issues exist, e.g. citation 12 is not about breast cancer but about prostate cancer; citation 13 used phantoms not mouse models. The authors should have carefully checked their citations as was already suggested last time. 

Manuscript readability is fine. 

Author Response

Reviewer 3:

The manuscript has shown improvements. However, several obvious issues exist, e.g. citation 12 is not about breast cancer but about prostate cancer; citation 13 used phantoms not mouse models. The authors should have carefully checked their citations as was already suggested last time. 

Answer: We thank the reviewer for the suggestions. Citation 13 has been corrected in the content, as recommended. As per Citation 12, it includes both breast and prostate cancer and more (renal, melanoma) in its content; in the context of our manuscript, it is used to substantiate the relationship between RT/IO in renal, pancreatic, and melanoma cancers. That’s why we have decided to include it.

This manuscript is a resubmission of an earlier submission. The following is a list of the peer review reports and author responses from that submission.

Round 1

Reviewer 1 Report

Combination therapy of radiation therapy and immunotherapy recently attracts more attention and PET imaging can contribute to this emerging research field.  Thus, this can be a timely commentary.  However, there is still much room for improvement in this manuscript.  The first and most important comment is that authors need to be more clear on what they want to discuss in this manuscript.  Do they want to review historical data of FDG PET/CT imaging for the combination therapy in human as the title suggested?  Do they want to propose guidelines for the application?  Do they want to promote general PET imaging as a tool to guide/monitor the combination therapy?  Is it just a mini review with some perspectives?  In the introduction, authors need to clearly depict what readers can expect from this manuscript.  It is quite confusing as of now.   Secondly, terms need to be unambiguous.  For example, in many occasions, authors use PET and FDG PET interchangeably.  However, authors also mentioned other PET tracers in the manuscript that causes confusion.  I listed a few representative examples below, but please edit/revise the manuscript thoroughly to avoid further confusion.  The other major comment is that Tables and Figures need to be more elaborated in the manuscript.  I do not know if they are necessary for this commentary without further explanation.  Overall, this is a timely and interesting manuscript.  I'd like to review it again after authors revise it more thoroughly.

Additional comments or suggestions:

1.  Many information in the manuscript is vague and not clear.  For example, authors mentioned a tight relationship between bone marrow glucose metabolism and myeloid infiltration in the tumor microenvironment, but did not describe what the relationship is.  Also, throughout the manuscript, when examples were given, more information needs to be provided including if an example is related to RT, IO, or RT/IO.  Also, correct terms have to be used to avoid any unnecessary confusion. 

2. Authors need to more elaborate on Tables and Figures.  For example, authors need to spend more space on the explanation of the proposed guidelines for FDG PET in Figure 1 to convince readers. 

3. Please make sure that all abbreviations are correct. I can see multiple errors (e.g. [18]Fluorodeoxyglucose --> [18F]-fluorodeoxyglucose, [89]Zr labeled PD-1 --> [89Zr] labeled anti-PD-1 antibodies, etc.

4. On P2, section 2.2. "the expression of checkpoint inhibitors (ICI)" does not make sense.  It is supposed to be "the expression of immune checkpoint proteins"?  Also, in many occasions, authors use terms incorrectly.  e.g. on p2, section 2.2. "tumor response to PD-1" is not correct.  PD-1 is protein expressed on the immune cells.  It should be PD-1 inhibitors.  I will not point out everything that needs to be corrected.  Please revise the manuscript thoroughly to avoid any further confusion.

5. Immuno-PET is generally considered as PET imaging with radiolabeled antibody.  PET Imaging of immune context can be done with other radiotracers including peptide- and small molecule-based ones as authors also included 18F-AraG in the immuno-PET section. 

Reviewer 2 Report

  1. Lack of extensive citations: The article lacks sufficient references to support the information presented, particularly in the sections discussing immuno-PET. As a review article, it is crucial to provide a comprehensive and evidence-based overview of the existing literature on the topic. I recommend adding more references to substantiate the claims made and enhance the scientific rigor of the article.

  2. Lack of clear distinction between review and scientific research: The article appears to be a mix of a review and a scientific research paper, which may not align with the typical scope of "Cancers" journal. While the article aims to summarize the current imaging guidelines, it lacks a clear research question or hypothesis, and does not present new scientific findings or original research data.

  3. Misalignment with the journal's scope: Based on the content and format of the article, it may not be closely related to the core focus of "Cancers," which primarily focuses on scientific research. I suggest considering other journals that are more suitable for this type of article, such as a clinical guideline publication or a review journal that specifically focuses on imaging guidelines.

In light of these points, I recommend revising and resubmitting your article to a more appropriate journal that aligns with the content and format of your work. I appreciate your efforts in preparing this submission and look forward to seeing your revised work in a more suitable venue.

Reviewer 3 Report

The content of this manuscript is of high interest and significance. The manuscript is concise and overall flow is easy to follow.

Areas that need attention include:

1)        Some of the specific contents/statements lack citations.

2)        Some of the specific information couldn’t be found from the citations, particular citations 10, 11, 12. It’s likely the wrong citations were inserted.

3)        As such, the authors should double check the correctness of all citations/references.

4)        Treatment response/results in the first column needs to be spelled out, such as CR for complete response, etc. because these abbreviations are not listed anywhere.

 More specific comments can be found as inserted comments.

Reviewer 4 Report

Manuscript number: cancers-2341501

Summary: The commentary entitled “[18F]FDG PET/CT Imaging in Patients with Solid Tumors Treated with Radiation Therapy Combined with Immunomodulatory Treatment” discussed current 18F-FDG PET/CT for defining the effect and refining patterns of response to IO/RT in patients with solid tumors treated with IO/RT, which offers a brief guideline diagnostic application of patients with solid tumors for IO/TR treatment.

1. Could the author briefly describe Figure 1 at the end of section 3?

2. 18F-FDG is widely used in clinical practice, but could the authors discuss the limitation of 18F-FDG imaging and perspective future development of molecular imaging for cancer patients?